# Water−Processed Organic Solar Cell with Efficiency Exceeding 11%

**DOI:** 10.3390/polym14194229

**Published:** 2022-10-09

**Authors:** Chen Xie, Songqiang Liang, Guangye Zhang, Shunpu Li

**Affiliations:** College of New Materials and New Energies, Shenzhen Technology University, Shenzhen 518118, China

**Keywords:** organic solar cells, surfactant−stripped nanoparticle, high efficiency, solvent additive

## Abstract

Water processing is an ideal strategy for the ecofriendly fabrication of organic photovoltaics (OPVs) and exhibits a strong market−driven demand. Here, we report a state−of−the−art active material, namely PM6:BTP−eC9, for the synthesis of water−borne nanoparticle (NP) dispersion towards ecofriendly OPV fabrication. The surfactant−stripping technique, combined with a poloxamer, facilitates purification and eliminates excess surfactant in water−dispersed organic semiconducting NPs. The introduction of 1,8−diiodooctane (DIO) for the synthesis of surfactant−stripped NP (ssNP) further promotes a percolated microstructure of the polymer and NFA in each ssNP, yielding water−processed OPVs with a record efficiency of over 11%. The use of an additive during water−borne ssNP synthesis is a promising strategy for morphology optimization in NP OPVs. It is believed that the findings in this work will engender more research interest and effort relating to water−processing in preparation of the industrial production of OPVs.

## 1. Introduction

A representative organic electronic, organic photovoltaic (OPV), is considered to be part of a new generation of photovoltaics devices with widespread application prospects due to its properties, including lightweight, semitransparency, high flexibility and low cost solution−processable fabrication [1,2]. Despite its efficiency reaching 19% [3,4], which far exceeds the threshold of 15% for the commercialization of OPVs [5,6], the hazard solvents [7] for device fabrication, such as chloroform (CF), chlorobenzene (CB), and 1,2−dichlorobenzene (DCB) [8], still restricts the large−scale production and widespread application of OPVs. Therefore, an environmentally compatible substitute for solution−processing, ideally water, is a prerequisite to realize mass production and the lab−to−fab conversion of OPV research.

One concept to combine hydrophobic light−harvesting materials with water is to insert hydrophilic pendants to the conjugated molecules. Those groups are ionic (such as ammonium or sulfonate) or non−ionic (such as quaternary ammonium salt, hydroxyl and phosphonate) [9]. Due to the charge−trapping effect and quenching sites caused by polar groups, water soluble active materials only exhibited low efficiencies [10,11,12,13,14]. Another concept to process state−of−the−art photovoltaic materials from water is to produce nanoparticle (NP) inks [15]. The NP strategy exhibits a fairly good ability to phase domain manipulation in bulk−heterojunction (BHJ) films as a result of a handle that precisely controls the particle size during the NP synthesis process [16].

NP inks with a high performance polymer/nonfullerene acceptor (NFA) system can be developed through emulsion or precipitation techniques [16,17]. However, surfactant is inevitable during the water−based NP synthesis process. The insulating and ionic characters of residual surfactant results in limited charge transport, which typically causes severe recombination in NP−based solar cells [18]. Therefore, the removal of excess surfactant was reported by many teams through washing steps, dialysis or centrifugal dialysis before film deposition, as well as water/alcohol treatment after film deposition [19,20,21]. However, the surfactants from those reports, such as ionic sodium dodecyl sulfate (SDS), alkyl trimethylammonium bromide (C_X_TAB) and nonionic alkylethoxyethyl (C_m_E_n_), prefer binding to the NP surface, resulting in insufficient surfactant reduction [20,22]. Our previous work introduced poloxamer as a surfactant and minimized the negative effect of surfactant on NP−based solar cell performance [15]. This surfactant−stripped NP (ssNP) strategy elevated the power conversion efficiency (PCE) record of the water−processed organic solar cells up to 7.5%. This approach introduces a generic concept for aqueous−processed OPV devices with promising high−efficiency and stability.

In this work, we combined a state−of−the−art polymer/NFA system, namely PM6:BTP−eC9 [23], with a poloxamer−based ssNP strategy. Owing to the excellent processability of multiple solvent systems, such as CF, toluene (Tol), tetrahydrofuran (THF) etc. [24], PM6 was a promising donor material to process NP−OPVs with high performance. A derivative of Y6 [25], BTP−eC9, also exhibits high tolerance on non−halogenated solvents, and thereby, undergoes reasonable molecular stacking and device efficiencies by means of “green” solvent processing [24]. We developed a PM6:BTP−eC9 light−harvesting blend into a frozen micelle state through an adapted nanoprecipitation technique [15,26]. The surfactant used in this process, poloxamer, was converted to non−micellar form at low temperatures, and was then virtually eliminated through a surfactant−stripping process, leaving behind almost surfactant−free NPs inks with extremely high purity. Moreover, the introduction of 1,8−diiodooctane (DIO) into the pristine THF solution promoted intimate intermixing among donors and acceptor molecules in the ssNP films. Ultimately, the aqueous ssNP−processed OPVs obtained a PCE of over 11%. This is the first time that water−processed NP solar cells surpassed a threshold efficiency of 10%. The following optical properties and the device’s physics investigation clarified the charge transport and recombination behaviors of the ssNP system with DIO as an additive. Therefore, a common strategy for water−processing OPVs with highly efficient light harvesting materials, which addresses the minimized impurity and percolated film morphology, is presented.

## 2. Materials and Methods

### 2.1. Materials

PM6 and BTP−eC9 were purchased from Solarmer Material Inc. (Beijing, China). Poloxamer 407 was purchased from Sigma Aldrich (St. Louise, MO, USA). Zinc oxide (ZnO) nanoparticles were purchased from Avantama AG (Stäfa, Switzerland). The other materials and solvents were commercially available and used as received.

### 2.2. NP Synthesis

The PM6:BTP−eC9 (1:1.2, *w*/*w*) mixture (4 mg) was dissolved in THF (1 mL) with 1% DIO as an additive. This solution was gently stirred for 3 h at 40 °C. A water solution of poloxamer was prepared using deionized water and the concentration was 20 mg mL^−1^. During the NP synthesis process, the THF solution was injected into the poloxamer solution at a ratio of 1:20 in a sonic bath. After sonification for 2 min, an A stream of N_2_ gas went through the solution to remove THF.

For the surfactant−stripping process, the NP dispersion with poloxamer was first cooled to 0 °C. The cooled NP system was then transferred into an Amicon^®^ ultra−15 centrifuge filter (Sigma Aldrich, St. Louise, MO, USA) (cutoff 100 K) and was centrifuged at 4000 rpm for 10 min at 4 °C. The retentate was raised to 15 mL with water and then centrifuged again. This process was repeated five times. The retentate was filtered through a 0.45 μm filter before the last centrifugation. The final ink was concentrated to 50 mg mL^−1^.

### 2.3. Poloxamer Quantification

A reported colorimetry assay [27], based on the capability of cobalt thiocyanate to form a complex with poloxamer polymers, was used to determine the concentration of poloxamer. In our protocol, 0.3 g cobalt nitrate hexahydrate and 1.2 g ammonium thiocyanate were dissolved in 3 mL of water for cobalt thiocyanate preparation. A 300 μL solution of cobalt thiocyanate was then combined with a 120 μL poloxamer solution with a concentration range from 1 to 20 mgmL^−1^, 600 μL ethyl acetate and 240 μL ethanol. After gently stirring, the mixture was centrifuged at 12,000 rpm for 5 min. The blue supernatant was then removed using a pipette and the blue pellet was left in the tube. The blue pellet was washed several times using ethyl acetate until the supernatant became colorless. The pellet was then redissolved in 1 mL of acetone to measure the absorbance at 623 nm. Appendix A shows the calibration curve of the absorbance of the cobalt thiocyanate−poloxamer complex as a function of the concentration of poloxamer. The poloxamer retention was obtained by collecting the filtrate after every washing step and by calculating the concentration with this curve. The retention of PM6:BTP−eC9 was determined by measuring the absorbance of retentate solution.

### 2.4. Characterization

Particle size and distribution were determined by dynamic light scattering (DLS) using a Malvern Zetasizer Nano ZS90 from Malvern Panalytical (Malvern, UK). UV/Vis absorption spectra were measured using an UV−Vis−NIR spectrometer (Lambda 1050, from Perkin Elmer, Waltham, MA, USA). Photoluminescence (PL) spectra were detected by a FLS1000 Spectrometer from Edinburgh Instruments (Livingston, UK). Scanning electron microscope (SEM) results were obtained from the field emission scanning electron microscopy (FESEM) GeminiSEM 300 (Carl Zeiss Microscopy Ltd., Jena, Germany). Transit Photovoltage (TPV) and Transit Photocurrent (TPC) were investigated by a TranPVC setup from Oriental Spectra Technology (Guangzhou, China) Co., Ltd.

### 2.5. Solar Cell Fabrication and Characterization

The solar cells were processed in an inverted architecture, namely glass/ITO/ZnO/active layer/MoO_x_/Ag. Indium tin oxide (ITO) substrates were cleaned by ultrasonication in acetone and isopropanol. Then, 30 nm ZnO (Avantama N−10) was doctor bladed on the ITO substrate and annealed at 85°C in air. Subsequently, the ssNP dispersion of PM6:BTP−eC9 (50 mg mL^−1^) was spin−coated at 1200 rpm onto an ZnO surface in ambient conditions and then stored in a vacuum overnight. After water elimination, the films were annealed at 100°C in a glovebox for 10 min. A total of 10 nm molybdenum oxide and 100 nm silver were subsequently deposited by thermal evaporation through a shadow mask to form an active area of 0.06 mm^2^. The current–voltage characteristics of the solar cells were measured under AM 1.5 G irradiation on a Newport solar simulator (Taiwan, China). The light source was calibrated using a silicon reference cell. All the cells were tested under an inert atmosphere. EQEs were measured using an Enlitech QE−S EQE system (Taiwan, China) that was equipped with a standard Si diode. Monochromatic light was generated from a Newport 300 W lamp source. The long−term photo−stability of the solar cells was performed under continuous one−sun illumination in a home−built chamber filled with N_2_.

### 2.6. Space−Charge−Limited−Current (SCLC)

Single carrier devices were fabricated and the dark J–V characteristics were measured and analyzed in the SCL regime following the previous literature. The architecture of the hole−only devices was glass/ITO/PEDOT:PSS (30 nm)/active layer/MoO_x_ (10 nm)/Ag (100 nm). The architecture of the electron−only devices was glass/ITO/ZnO (30 nm)/active layer/Ca (5 nm)/Ag (100 nm). The reported mobility data are the average values over the 18 devices of each sample for a range of thicknesses. The SCLC curves can be fitted to the Mott–Gurney relation for SCLC [28] as follows
(1)$$J_{SCL}=\frac{9}{8}\varepsilon_{0}\varepsilon_{r}\mu \frac{V_{in}^{2}}{L^{3}}\exp\left(\frac{0.89\beta }{L^{0.5}}V_{in}^{0.5}\right)$$
where *J*_SCL_ is the current density, *ε*_0_*ε*_r_ is the dielectric permittivity, *µ* is the carrier mobility, *L* is the film device, and *β* is the field activation factor [16].

## 3. Results and Discussions

### 3.1. Syhthesis of Surfactant−Stripped NPs

As shown in Figure 1a, an amphiphilic polymer poloxamer, with an ethylene oxide backbone, was selected as a nonionic surfactant for NP synthesis because of its temperature sensitive critical micelle concentration (CMC) properties [29,30]. Figure 1b,c demonstrate the nanoprecipitation approach for ssNP synthesis through a CMC−switching strategy [26]. Light−harvesting blends, PM6 and BTP−eC9, were first dissolved in THF. The solutions were injected into a water solution of poloxamer under sonication. The sudden drop in solubility caused precipitation of the hydrophobic conjugated semiconductors and finally formed an NP dispersion [31,32]. The CMC−switching, manifesting a sharp rise in CMC, induced a conversion of poloxamer micelles back to free dispersed molecules in water by lowering the temperature of the NP dispersion to around 4 °C. Thus, free and loosely bound poloxamers were readily removed through the centrifugal filtration (described in the experimental process), leaving behind retentate with relatively pure NP ink. This property of temperature−sensitive CMC makes poloxamer a unique type of surfactant in comparison to other surfactants. The ssNP ink not only minimizes the negative impact of the surfactant on the NP solar cells, but restrains the occurrence of aggregates in surfactant−free NP systems [33].

Figure 2a shows the retention of poloxamer in the ssNP system during the surfactant−stripping process. The filtrate obtained after each centrifugal washing step was collected to evaluate the quantity of residual poloxamer via a colorimetry assay (as described in the experimental process). The retention values were obtained from the absorption of cobalt thiocyanate–poloxamer complexes from each filtrate in Appendix A. It is obvious that a fivefold stripping process eliminated most excess poloxamers, while the quantity of light−harvesting material was unchanged and did not lead to sedimentation. Figure 2b,c exhibit well−dispersed spherical ssNPs without aggregated clusters, even after the stripping of most surfactants. The size of the NPs determines the phase separation and intermixing of the donor and acceptor in the NP films. It was reported in our previous work that, by precisely controlling the size of the pristine NPs in the nm regime, it is possible to generate thin films with optimized domain sizes [16]. An average diameter of 90 nm was suitable for the deposition of an active layer film with a thickness of approximately 100 nm. This is due to the deformation of the organic semiconductor NPs during spin−coating and the drying process [15]. As shown in Appendix A, thermal annealing−evolved ssNP films on the ZnO surface formed a discrete structure with isolated spheres to an interconnected one with an ambiguous surface, which is beneficial for charge transport through the percolated NP film [34].

### 3.2. Solar Cell Characterization

We investigated ssNP inks for solar cell fabrication in an inverted architecture (Figure 3a,b). The water dispersion was deposited on top of the ZnO NP film by spin−coating. In the PM6:BTP−eC9 film deposited with different solvents, the normalized absorption peaks (λ_max_) of PM6 and BTP−eC9 in the ssNP film demonstrated a red−shift from 627 to 635 nm and 826 to 831 nm, in comparison to those in the solution−processed BHJ film, respectively (Figure 3c). A further red−shift occurred in the ssNP system without the addition of DIO during the NP synthesis process (λ_max_ of 636 and 836 nm for PM6 and BTP−eC9, respectively), which indicated a conversion in the molecular packing behavior. Compared to DIO−free ssNPs, a lesser red−shift suggests less aggregates and phase−separation in the ssNP film with DIO [35]. The PL spectra of DIO−free ssNP films represents an incomplete PL quenching at both PM6 and BTP−eC9 peaks, thus further confirming an insufficient percolation of donor and acceptor without an additive (Figure 4). A similar phenomenon of a red−shift was also observed in the absorption spectra of DIO−free ssNP water dispersions (Appendix A), thus suggesting that the DIO additive could retard the phase−separation process during ssNP synthesis. Our previous work indicated that the fixed microstructure of each NP determines the ultimate film morphology in the NP solar cells [16]. Thus, the addition of DIO in the THF solution of PM6:BTP−eC9 affects the microstructure of NPs and ultimately has an impact on solar cell performance.

The introduction of DIO in the ssNP synthesis also has a significant impact on solar cell performance. As shown in Figure 3d and Table 1, the solar cell processed by DIO−free ssNP during synthesis exhibits a V_OC_ of 0.72 V, a J_SC_ of 18.9 mA cm^−2^, and an FF of 0.66, resulting in the best PCE of 9.04%. When the ssNP system has the addition of DIO, the values of V_OC_, J_SC_, and FF increased to 0.78 V, 20.7 mA cm^−2^, and 0.69, respectively. A champion efficiency of 11.1% was obtained. It should be noted that this was the first time that a highly efficient PM6 system was introduced to the ssNP solar cell and both types of devices broke the PCE record for water−processed NP OPVs. A pronounced external quantum efficiency (EQE) in the range of near infrared is the primary cause of high J_SC_ in devices with DIO. The J_SC_ integration of EQE curves is also shown in Figure 3e. This delivered values of 18.5 and 20.6 mA cm^−2^ for solar cells without and with DIO, respectively, and fitting with an error of less than 0.5 mA cm^−2^ to the AM1.5 simulated measurements. Appendix A shows the long−term operational stability of devices under a maximum power point (MPP) tracking in an N_2_ atmosphere. The DIO−free and DIO−contained solar cells exhibited PCE losses of 32% and 24% after 2000 h lifetime testing under 1 sun illumination, respectively. The ssNP solar cells with DIO presented PCE losses of 20% after 1300 h irritation, which was comparable to that of our previous reported PM6:IT−4F device [36], thus suggesting that the water−processing did not negatively influence the intrinsic stability of the PM6−based solar cells.

### 3.3. Device Physics Analysis

To study the origin of the high J_SC_ value and inferior V_OC_ loss in the PM6:BTP−eC9−based ssNP solar cell with DIO, the charge recombination behavior was first investigated with light intensity−dependent J–V measurements (Figure 5 and Appendix A). The nongeminate recombination behavior was investigated with J_SC_ as a function of light intensity (P_in_). As shown in Figure 5a, using the power law dependence of J_SC_ upon light intensity, the fitted slopes for both types of solar cells were around one, thus indicating that the J_SC_ reduction of the device without DIO was not mainly caused by bimolecular recombination [37,38]. Thus, first−order recombination caused by trapping or inefficient charge separation would be the dominate reason for inferior J_SC_’s in DIO−free ssNP solar cells. In addition, the V_OC_ slope versus ln (P_in_) for the device with DIO was much closer to kT/q than that of the device without DIO (1.44 kT/q for the device without DIO and 1.27 kT/q for the device with DIO), thus suggesting that the DIO−free device suffered from critical trap−assisted first−order recombination in open−circuit conditions. This also illustrates a stronger V_OC_ loss in the DIO−free device. Therefore, the DIO−contained ssNP solar cell was expected to have less trapping states and more efficient charge generation [39,40,41].

The charge−carrier mobilities of PM6:BTP−eC9−based ssNP solar cells were tested using the SCLC model through fabrication of a hole−only and electron−only device with ssNPs (see **experimental process**). As shown in Figure 6a, the DIO−free device exhibited a relatively low hole mobility of 8.88 × 10^−5^ cm^2^ V^−1^ s^−1^, as compared to that of the device with DIO (1.54 × 10^−4^ cm^2^ V^−1^ s^−1^), which has almost twofold the magnitude of the DIO−free device. Nevertheless, the values of the electron mobility in these two types of devices are much closer, which exhibits a μ_e_ of 1.32 × 10^−4^ cm^2^ V^−1^ s^−1^ and 1.41 × 10^−4^ cm^2^ V^−1^ s^−1^ for solar cells without and with DIO, respectively. It is suggested that the imbalanced μ_h_ and μ_e_ mobilities limit the charge extraction in DIO−free devices and results in restrained J_SC_ and FF with severe bimolecular recombination (Table 1) [16]. The SCLC results further reveal that the performance loss in the DIO−free device is partially caused by inefficient and imbalanced charge transport.

To further probe the charge transport and collection behaviors in ssNP solar cells, the TPV and TPC were measured to evaluate the impact of DIO on the charge carrier lifetime (τ) and sweeping out time (t_S_) of the devices [42,43]. As shown in Figure 7, the devices with DIO exhibits a longer τ value of 1.43 μs and shorten t_S_ value of 0.118 μs in comparison with those of DIO−free device (τ = 1.24 μs and t_S_ = 0.119 μs). The prolonged lifetime of the carriers and the accelerated charge collection would lead to enhancements of J_SC_ and FF values in in solar cells with DIO as depicted above.

## 4. Conclusions

In conclusion, we have introduced poloxamer as a surfactant−stripped stabilizer for synthesis of water−dispersed NPs for eco−friendly OPV processing. One of the state−of−the−art light−harvesting polymer/NFA combination, PM6:BTP−eC9, was processed into a surfactant−stripped NP system with minimized residual surfactant and high purity. The PM6−based solar cells were fabricated from an aqueous NP ink and finally delivered a PCE over 11%, which was the first time for a water−processed OPV with an efficiency over 10%. The addition of DIO to precursor THF solution for NP synthesis plays a crucial role on the PCE enhancements in PM6:BTP−eC9−based ssNP OPVs. It was found that morphology difference occurs when addition of DIO during the NP synthesis process before NP film deposition. Optical property measurements such as absorption and PL quenching demonstrate insufficient phase−separation and incomplete percolation in DIO−free ssNP films. V_OC_ dependence upon light intensity indicates that trap states result in severe V_OC_ loss in DIO−free device. SCLC shows inefficient charge transport and pronounced charge recombination from solar cells without DIO. TPV and TPC results further confirm less recombination and efficient collection of charge in the optimized solar cells with DIO. It is the first time showing the significance of additive on organic semiconductor NP formation and performance of NP OPVs. Introduction of additive for NP synthesis allows an improved microstructure in each NP as well as solar cells. This study has pointed out one of the routines to further enhance the performance of NP OPVs. It is nevertheless believed that aqueous organic NP inks provide an inspiration for OPV production towards ecofriendly processing.

## Figures and Tables

**Figure 1 polymers-14-04229-f001:**
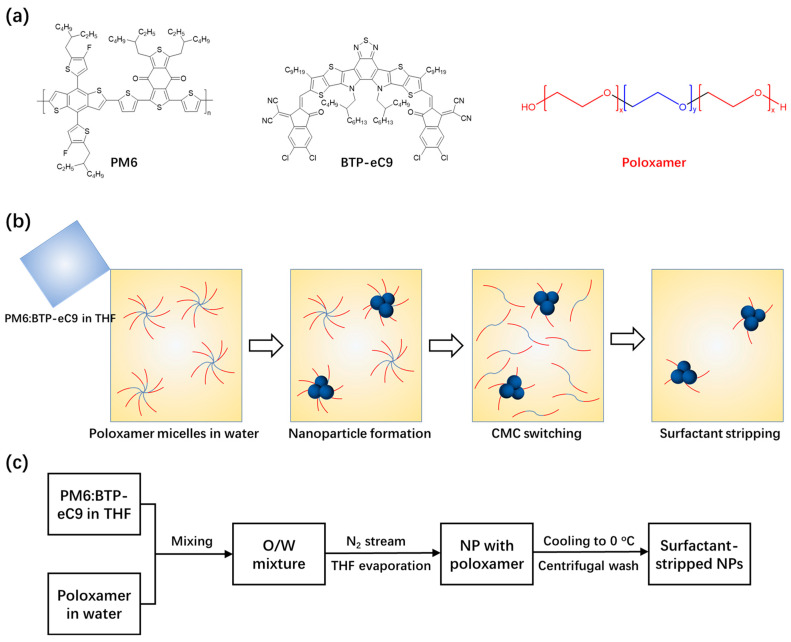
(**a**) Chemical structure of donor PM6, acceptor BTP−eC9, and surfactant poloxamer for ssNP synthesis; (**b**) synthesis and purification of ssNPs by temperature−mediated CMC−switching and the surfactant stripping strategy; and (**c**) flowchart of the ssNP synthesis.

**Figure 2 polymers-14-04229-f002:**
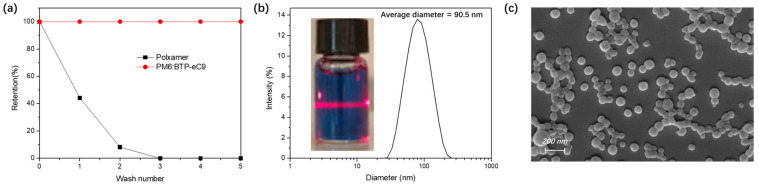
(**a**) Poloxamer retention as a function of centrifugal washes at 4 °C in aqueous PM6:BTP−eC9 ssNP ink; (**b**) size distribution of PM6:BTP−eC9 ssNP ink and its tyndall effect; and (**c**) SEM images of dried PM6:BTP−eC9 ssNP ink spin−coated on the silicon slide. The scale bar was 200 nm.

**Figure 3 polymers-14-04229-f003:**
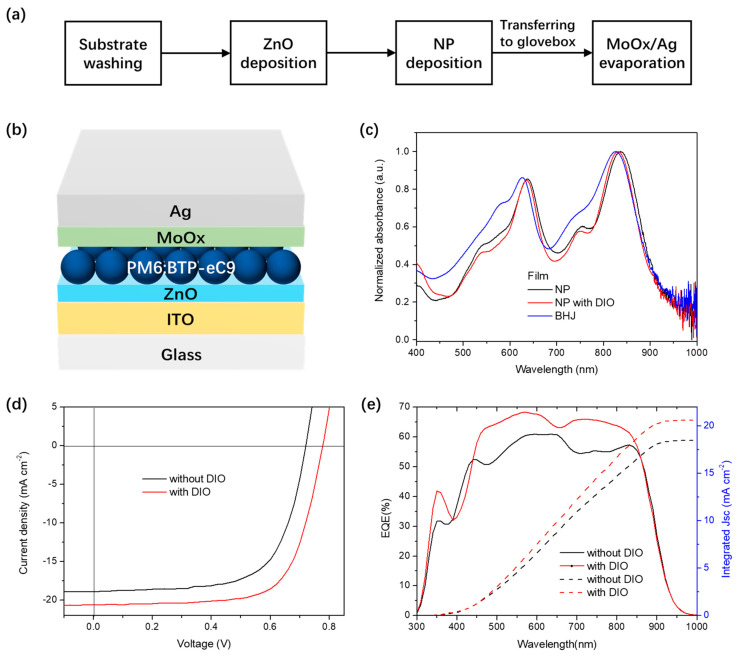
(**a**) Flowchart of PM6:BTP−eC9 ssNP solar cell fabrication; (**b**) device architecture of the PM6:BTP−eC9 ssNP solar cell; (**c**) normalized UV−vis absorption spectra of PM6:BTP−eC9 ssNP and chloroform solution−processed films; (**d**) J−V characteristics; and (**e**) EQE of PM6:BTP−eC9 ssNP processed solar cells with or without DIO.

**Figure 4 polymers-14-04229-f004:**
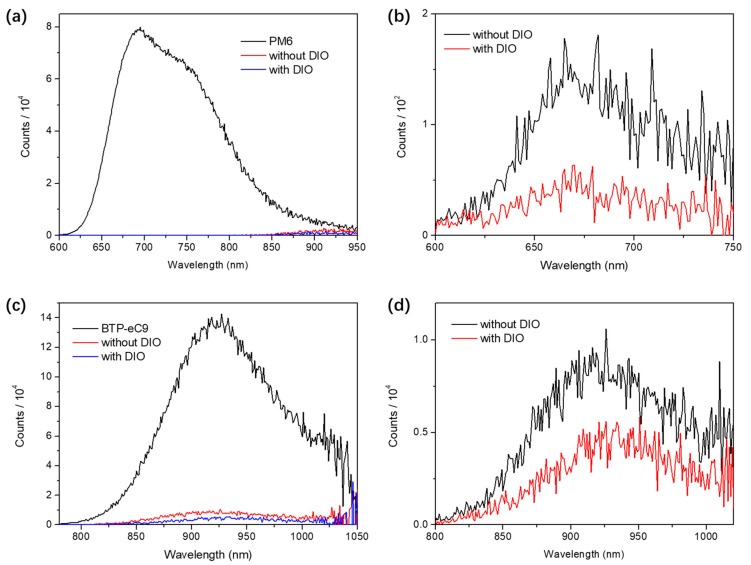
PL spectra of neat PM6, neat BTP−eC9, and ssNP films under excitation of (**a**,**b**) 560 nm and (**c**,**d**) 750 nm, respectively.

**Figure 5 polymers-14-04229-f005:**
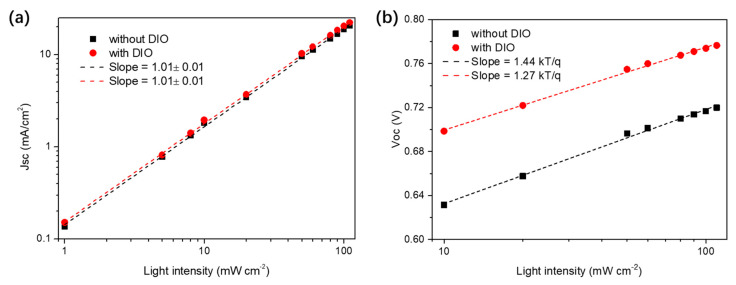
(**a**) J_SC_ and (**b**) V_OC_ as a function of light intensity for ssNP solar cells with or without DIO.

**Figure 6 polymers-14-04229-f006:**
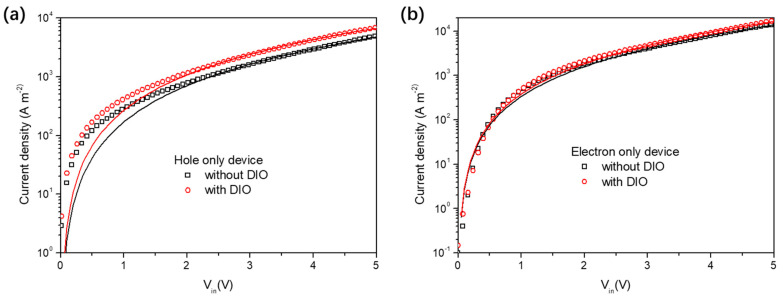
Dark J–V characteristics of PM6:BTP−eC9−based (**a**) hole−only and (**b**) electron−only ssNP devices. Hole−only device structure: glass/ITO/PEDOT:PSS/NPs/MoO_x_/Ag; electron−only device structure: glass/ITO/ZnO/NPs/Ca/Ag. The black and red lines were the fitted curves from the Mott–Gurney relation (see Section 2.6).

**Figure 7 polymers-14-04229-f007:**
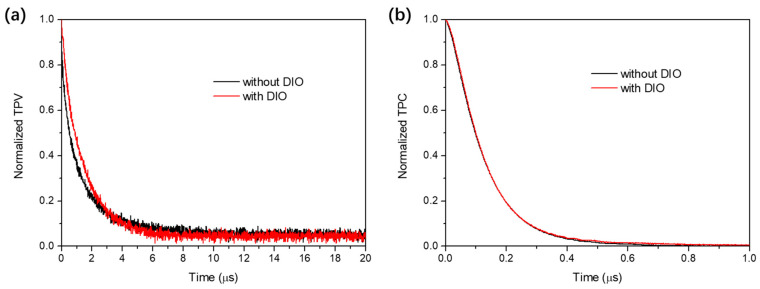
Normalized (**a**) TPV and (**b**) TPC curves of PM6:BTP−eC9−based ssNP devices with or without DIO.

**Table 1 polymers-14-04229-t001:** Summary of the photovoltaic parameters of the PM6:BTP−eC9−based ssNP solar cells with and without DIO.

Active Layer	V_OC_ (V)	J_SC_ ^(a)^ (mA cm^−2^)	J_SC_ ^(b)^ (mA cm^−2^)	FF	PCE ^(c)^ (%)	PCE ^(d)^ (%)	μ_h_/μ_e_	τ (μs)	t_S_ (μs)
without DIO	0.72 ± 0.2	18.2 ± 0.5	18.5	0.65 ± 0.03	8.52 ± 0.52	9.04	0.67	1.24	0.119
with DIO	0.77 ± 0.2	20.4 ± 0.8	20.6	0.68 ± 0.05	10.70 ± 0.42	11.1	1.09	1.43	0.118

^(a)^ Average J_SC_ values obtained from the solar simulator. ^(b)^ J_SC_ values calculated from the EQEs of devices with the best PCEs. ^(c)^ Average PCE values and ^(d)^ Best PCE values.

## Data Availability

The data presented in this study are available on request from the corresponding author.

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
