# Peer review of "Water−Processed Organic Solar Cell with Efficiency Exceeding 11%"

_polymers, 2022, doi:10.3390/polym14194229_

Round 1
Reviewer 1 Report
The manuscript should be published after minor revision
The authors report active materials PM6:BTP-eC9 for synthesis of water-borne nanoparticle (NP) dispersion and ecofriendly OPV fabrication. They combind the surfactant-stripping technique combined with a poloxamer and the introduction of 1,8-diiodooctane (DIO) to the synthesis of surfactant-striped NP (ssNP). Finally, authors yielded water-processed OPVs with a record efficiency over 11%.
The paper is presented in a concise and clear way, the article is well written with care and perfection but I have some comments:
Authors talk about green solvents in the line 57. Authors should remove the reference 21 due to authors use non-green solvents in this reference.
The authors should remove the word eco-friendly from the conclusions and abstract. The OPV process preparation in this paper is not totally eco-friendly. Authors should include the word “towards” before eco-friendly
Authors should include the name of the authors and the affiliations in the Supporting Information
Author Response
Response to Reviewer 1 Comments
Point 1: Authors talk about green solvents in the line 57. Authors should remove the reference 21 due to authors use non-green solvents in this reference.
Response 1: T hank you for your comment. The reference 21 has been deleted.
Point 2: The authors should remove the word eco-friendly from the conclusions and abstract. The OPV process preparation in this paper is not totally eco-friendly. Authors should include the word “towards” before eco-friendly.
Response 2: Thank you for your comment. As you mentioned, the word ”ecofriendly” in “Keywords” has been deleted. The sentences with “ecofriendly” in abstrate and conclusion have been revised and the word”towards” has been added before “ecofriendly”. Plesase check the revised manuscript.
Point 3: Authors should include the name of the authors and the affiliations in the Supporting Information.
Response 3: Thank you for your comment. Please see the revised Supporting Information. The name of the authors and the afflilications have been added to the manuscript.
Reviewer 3 Report
This study reports poloxamer as a surfactant-stripped stabilizer for synthesizing water-dispersed NPs for eco-friendly OPV processing. The work is suitable for publication after addressing the following issues.
1. In the introduction, more details are required on surfactant-stripped NPs systems. Authors need to report more studies and compare how this work is superior to previous studies.
2. Authors report that 90 nm diameter of ssNPs is suitable. Authors should briefly discuss the effect of changing diameters on the overall performance.
3. Authors are invited to report the stability of the proposed device and compare it with the existing literature.
